# CXCL1/CXCR2 Paracrine Axis Contributes to Lung Metastasis in Osteosarcoma

**DOI:** 10.3390/cancers12020459

**Published:** 2020-02-17

**Authors:** Chia-Chia Chao, Chiang-Wen Lee, Tsung-Ming Chang, Po-Chun Chen, Ju-Fang Liu

**Affiliations:** 1Department of Respiratory Therapy, Fu Jen Catholic University, New Taipei City 24205, Taiwan; 095457@mail.fju.edu.tw; 2Department of Orthopaedic Surgery, Chang Gung Memorial Hospital, Puzi City, Chiayi County 61363, Taiwan; cwlee@mail.cgust.edu.tw; 3Department of Nursing, Division of Basic Medical Sciences, and Chronic Diseases and Health Promotion Research Center, Chang Gung University of Science and Technology, Puzi City, Chiayi County 61363, Taiwan; 4Research Center for Industry of Human Ecology and Research Center for Chinese Herbal Medicine, Chang Gung University of Science and Technology, Guishan Dist., Taoyuan City 33303, Taiwan; 5School of Medicine, Institute of Physiology, National Yang-Ming University, Taipei City 11221, Taiwan; briancoinage@ym.edu.tw; 6Translational medicine Center, Shin-Kong Wu Ho-Su Memorial Hospital, Taipei City 11101, Taiwan; blibra1002@hotmail.com; 7Department of Biotechnology, College of Medical and Health Science, Asia University, Taichung 41354, Taiwan; 8Department of Medical Research, China Medical University Hospital, China Medical University, Taichung 40402, Taiwan; 9School of Oral Hygiene, College of Oral Medicine, Taipei Medical University, Taipei 11031, Taiwan

**Keywords:** Osteosarcoma, CXCL1, CXCR2, migration, VCAM-1

## Abstract

Osteosarcoma, the most common of all bone malignancies, has a high likelihood of lung metastasis. Up until now, the molecular mechanisms involved in osteosarcomas with lung metastases are not clearly understood. Recent observations have shown that the chemokine CXCL1 and its receptor CXCR2 assist with the homing of neutrophils into the tumor microenvironment. Here, we show that the CXCL1/CXCR2 paracrine axis is crucial for lung metastasis in osteosarcoma. In an in vivo lung metastasis model of osteosarcoma, lung blood vessels expressed CXCL1 and osteosarcoma cells expressed the CXCR2 receptor. CXCR2 expression was higher in osteosarcoma cell lines than in normal osteoblast cells. Immunohistochemistry staining of clinical osteosarcoma specimens revealed positive correlations between CXCR2 expression and pathology stage and also vascular cell adhesion molecule 1 (VCAM-1) expression. High levels of CXCL1 secreted by human pulmonary artery endothelial cells (HPAECs) promoted osteosarcoma cell mobility, which was mediated by the upregulation of VCAM-1 expression. When HPAECs-conditioned media was incubated in osteosarcoma cells, we observed that the CXCR2 receptor and FAK/PI_3_K/Akt/NF-κB signaling cascade were required for VCAM-1 expression. Our findings illustrate a molecular mechanism of lung metastasis in osteosarcoma and indicate that CXCL1/CXCR2 is worth targeting in treatment schemas.

## 1. Introduction

Of all bone malignancy diagnoses in children and young adults, osteosarcoma is the most common and frequently presents with metastasis at diagnosis, which contributes to mortality [1]. Approximately one-third of patients presenting with localized disease will relapse, as will about three-quarters of patients with metastases at diagnosis. As many as 90% of these relapses are due to lung metastases [2]. Therapy that effectively prevents osteosarcoma lung metastasis is urgently required.

Each type of carcinoma has its preferred sites of metastasis. Organs involved in metastasis secrete attractant molecules that guide cancer cells to specific sites [3]. It is also known that chemokines, low molecular chemotactic cytokines that mediate connection between different cell types, help to regulate leukocyte homing besides promoting cancer growth [4]. Indeed, chemokines play important roles in various biological functions including cell migration, angiogenesis, and hematopoietic cell homing [5]. They also play crucial roles in the progression and metastasis of different cancers. In breast cancer, the CXCL12/CXCR4 axis is implicated in the homing of cancer cells to metastatic sites [6,7]. Metastatic breast cancer cells express high levels of CXCR4 and stromal cells in distant organs express high levels of CXCL12, the CXCR4 ligand. The CXCL12/CXCR4 axis compromises adjuvant therapy in breast cancer [8].

Since it was first identified in culture supernatants of melanoma cell lines, CXCL1 has been implicated in inflammation, angiogenesis, tumorigenesis, and wound healing [9]. CXCL1 binds to CXCR2, which is highly expressed on neutrophil surfaces [10]. Tumor-derived CXCL1 modulates the tumor microenvironment by regulating various cells, such as macrophages, fibroblasts [11], neutrophils [12], and osteoclasts [13]. Increased levels of CXCL1 are associated with tumor size, advancing stage, depth of invasion, and patient survival [14,15,16]. Silencing of CXCL1 can inhibit tumor growth in hepatocellular carcinoma [17], while knockdown of CXCL1 expression can inhibit tumor growth in colorectal cancer liver metastasis [18]. CXCL1 autocrine and paracrine networks can also promote tumor invasion and metastasis [19,20,21,22]. However, although the pro-metastatic functions of CXCL1 are recognized in tumor progression, its role remains unclear in osteosarcoma. 

This report is the first to demonstrate that human pulmonary artery endothelial cells secrete CXCL1 and contribute to lung metastasis by increasing VCAM-1 expression. It also shows that CXCL1-promoted VCAM-1 expression is regulated by the CXCR2/FAK/PI_3_K/Akt/NF-κB pathway. Our results show that CXCL1 plays a pivotal role in metastatic osteosarcoma.

## 2. Materials and Methods

### 2.1. Materials

Anti-mouse and anti-rabbit IgG-conjugated horseradish peroxidase, rabbit polyclonal antibodies specific for CXCL1, CXCR2, VCAM-1, p-FAK, FAK, p85α, p-p85α, Akt, p-Akt, p-IKKα/β, IKKα/β, p-IκBα, IκBα, p-p65, p65, and β-Actin were purchased from GeneTex International Corporation (Hsinchu City, Taiwan). Recombinant human CXCL1 was purchased from PeproTech (Rocky Hill, NJ, USA). Short hairpin RNA (shRNA) plasmid for knocking down gene expression was purchased from the National RNAi Core Facility Platform (Taipei, Taiwan). All siRNAs were ON-TARGETplus siRNAs, purchased from Dharmacon Research (Lafayette, CO, USA). All other chemicals were obtained from Sigma-Aldrich (St. Louis, MO, USA).

### 2.2. Cell Culture

Human osteosarcoma cell lines (MG63, U2OS, and HOS) and normal osteoblast cell lines (hFOB1.19) were obtained from the American Type Cell Culture Collection (Manassas, VA, USA). Human pulmonary artery endothelial cells (HPAECs) were purchased from Lonza Clonetics (Walkersville, MD, USA). hFOB1.19 cells were cultured in DMEM/F12 medium, U2OS cells were cultured in McCoy’s 5A medium and MG63 and HOS cells were cultured in Eagle’s minimum essential medium. All cells were supplemented with contained with 20 mM HEPES, 10% heat-inactivated fetal bovine serum, 2 mM-glutamine, penicillin (100 U/mL), and streptomycin (100 μg/mL) at 37 °C with 5% CO_2_. The HPAECs were maintained in EGM^TM^-2 endothelial cell growth medium-2 BulletKit (Lonza), which contains basic growth medium (EBM-2), fetal bovine serum (FBS) and antibiotics, ascorbic acid, vascular endothelial growth factor (VEGF), human fibroblast growth factor (hFGF-B), hydrocortisone, human epidermal growth factor (hEGF), R3-IGF (insulin-like growth factor)-1, and heparin. Unless otherwise stated, cells were maintained in a 37 °C incubator with 5% CO_2_. Migration-prone MG63 cells were selected according to their differential migration ability; the cell culture insert system was used as described earlier [23]. After 24 h of migration, cells that penetrated through pores and migrated to the underside of the filters were trypsinized and harvested for a second selection process. Any of the original cells that did not pass through membrane pores were designated as M0. After 10 selection rounds, the migration-prone subline was designated as M10. After 20 selection rounds, the migration-prone subline was designated as M20.

### 2.3. Preparation of Conditioned Media

The HPAECs were seeded and grown overnight in 10-cm culture dishes. The cells were washed with PBS and cultured in EGM-2 medium for 48 h. The culture supernatants (conditioned media (CM)) were collected. To normalize for differences in cell density due to proliferation, the cells from each plate were collected, and the total DNA content per dish was determined (spectrophotometric absorbance, 260 nm).

### 2.4. Western Blot

The cells were lysed in RIPA lysis buffer and total cell lysates were collected. Resolved proteins were determined with SDS-PAGE and transferred to Immobilon™ polyvinyldifluoride (PVDF) membranes. Blots were blocked with 5% nonfat milk for 1 h at room temperature then incubated with primary antibodies for another 1 h at room temperature. After three washes, blots were incubated with peroxidase-conjugated secondary antibody (1:5000) for 1 h at room temperature. The blots were visualized using a charge-coupled device camera-based detection system (UVP Inc., Upland, CA, USA). Quantitative data were obtained using ImageJ software (National Institutes of Health, USA). All original western blot figures can be found in the Appendix A.

### 2.5. RNA Extraction and Quantitative Real-Time Polymerase Chain Reaction

Total RNA was extracted from cells using Total RNA preparation kits (easy-Blue Total RNA Extraction kit) purchased from iNtRON Biotechnology (Seongnam, Korea), following the manufacturer’s protocol. In brief, cells were added to 0.5 mL easy-Blue reagent, homogenized and incubated at room temperature for 3 min. After extraction with chloroform (0.1 mL) and precipitation with isopropanol (0.4 mL), RNA was washed with 75% ethanol, and then the RNA pellet was dissolved in 10 µL of RNase-free water. RNA yield and purity were determined by measuring absorbance at 260 and 280 nm using a Nanodrop spectrophotometer (Thermo Fisher Scientific, Inc., Waltham, MA, USA). Complementary DNA (cDNA) was generated by reverse transcription of total RNA, according to the manufacturer’s instructions (Invitrogen, Carlsbad, CA, USA).

Real-time quantitative polymerase chain reaction (qPCR) was performed using SYBR Green (KAPA Biosystems, Woburn, MA, USA), according to the manufacturer’s protocol, and reactions were performed using a StepOnePlus machine (Applied Biosystems, Foster City, CA, USA). Human VCAM-1 and glyceraldehyde 3-phosphate dehydrogenase (GAPDH) purchased from Sigma-Aldrich were used as primers to amplify the target genes. The expression levels of the target genes were determined by normalization to GAPDH levels. We calculated the results using this equation: Ratio = 2^−ΔΔCt^, where ΔΔCt = (Ct _target_−Ct _GADPH_)_Sample_−(Ct _target_−Ct _GADPH_)_Control_. Each sample was assayed in triplicate, and the data represent three independent experiments.

### 2.6. Transwell Cell Migration Assay

Cell migration assays were performed with Transwell inserts (8-μm pore size; Costar, NY, USA) in 24-well dishes. Cells (2 × 10^4^ cell/well) were pretreated for 90 min with the designated inhibitor then incubated with the culture supernatants for 24 h. The cells were seeded in the upper Transwell chamber, and 300 μL of medium was placed in the lower chamber. After 24 h, the cells were fixed in 3.7% formaldehyde for 30 min and stained with 0.05% crystal violet for 60 min. Cells on the upper side of the chamber were removed with cotton-tipped swabs, and the chamber were washed with PBS. Cells on the underside of the filters were examined and counted under a microscope. Each experiment was repeated at least three times.

### 2.7. Immunofluorescence Microscopy

MG63 cells (5 × 10^3^ cell/well) were seeded on glass coverslips and treated under the indicated conditions, rinsed once with PBS, and fixed in 3.7% paraformaldehyde for 15 min at room temperature. Cells were washed three times with PBS and blocked with 4% BSA for 15 min. After blocking, the cells were incubated with anti-human p65 (1:100) for 1 h at room temperature. After undergoing three washes with PBS, the cells were incubated with FITC-conjugated goat anti-rabbit IgG for 1 h. Finally, the cells were washed, mounted, and photographed using the Leica TCS SP2 Spectral Confocal System.

### 2.8. Chromatin Immunoprecipitation Assay

Chromatin immunoprecipitation (ChIP) analysis was performed as described previously [24]. DNA were extracted from treated samples and immunoprecipitated by anti-p65 antibody. The immunoprecipitated DNA were further subjected to purification by using phenol–chloroform. The purified DNA pellet was subjected to PCR then resolved using 1.5% agarose gel electrophoresis and visualized by ultraviolet illumination. Primers 5′-ACAGAGAGAGGAGCTTCAGCAGTGAGAGCA-3′ and 5′-GTCTGTGCTTTATAAAGGGTCTTGTTGCAG-3′ were used to amplify across the human promoter region of the NF-κB region (-2167 and -1967) in the promoter region of VCAM-1. 

### 2.9. Reporter Assay

The NF-κB report plasmid, pSV-β-galactosidase vector and luciferase assay kit were purchased from Promega (Madison, WI, USA). Cells (2 × 10^5^ cell/well) were co-transfected with NF-κB report plasmid and pSV-β-galactosidase vector for 24 h using Lipofectamine 3000™ (Invitrogen). Transactivation was determined by monitoring firefly luciferase levels in the pGL2 vector. The luciferase assay was performed by adding lysis buffer (100 μL), and cells were harvested by centrifugation (13,200 rpm for 5 min). The supernatant was transferred to fresh tubes, and 20 μL of cell lysate was added to 80 μL of fresh luciferase assay buffer in an assay tube. Luciferase activity was measured using a microplate luminometer and normalized to transfection efficiency based on the cotransfected β-galactosidase expression vector.

### 2.10. In Vivo Tumor Xenograft Study

All mice were obtained from the Lasco (Taipei, Taiwan). Ethical approval was obtained for the use of the animals, and all experiments were performed in accordance with the Guidelines for Animal Care of the Institutional Animal Care and Use Committee of College of Medicine, National Taiwan University (Approval No: 20150357). All experiments were performed in National Taiwan University. Four-week-old male CB17/SCID mice were purchased from Lasco and maintained under pathogen-free conditions. Seven animals per group were used, and the experiment was repeated twice. For assessing lung metastasis in osteosarcoma cells in the in vivo xenograft model. Two times 1 × 10^6^ cells were resuspended in 0.1 mL of PBS and injected into the lateral tail vein. After 4 weeks, mice were sacrificed using CO_2_. The lungs were removed and fixed in 10% formalin. The number of lung tumor nodules was counted using a dissecting microscope.

### 2.11. Immunohistochemistry (IHC) Staining

For investigating CXCR2 and VCAM-1 expression in clinical specimens, human osteosarcoma tissue arrays (BO244, T261, T262, T262A, T263, and OS804b) containing 11 cases of normal bone, 7 cases of stage I osteosarcoma, 49 cases of stage II osteosarcoma, and 7 cases of stage III osteosarcoma were purchased from Biomax (Rockville, MD, USA). The tissues were rehydrated, and incubated in 3% hydrogen peroxide to block endogenous peroxidase activity. After antigen retrieval, the sections were in 3% bovine serum albumin then incubated with the primary mouse polyclonal anti-CXCR2 and VCAM-1 antibody at 1:100 dilutions, at 4 °C overnight. After undergoing three PBS washes, samples were incubated with a 1:100 dilution of biotin-labeled goat anti-mouse IgG secondary antibody. Bound antibodies were detected using the ABC Kit (Vector Laboratories, Burlingame, CA, USA). Slides were stained with the chromogen diaminobenzidine, washed, counterstained with Delafield’s hematoxylin, dehydrated, treated with xylene, and then mounted. 

Stained specimens were photographed by microscope. Tumor cell staining intensities were scored from 0–5, where 0 = no staining or unspecific staining, 1 = very weak (intensity), 2 = weak staining, 3 = moderate staining, 4 = strong staining, and 5 = very strong staining. A pathologist scored staining intensity in all samples.

### 2.12. Statistical Analysis

Data are presented as the mean ± standard deviation (S.D). Statistical comparisons between two samples were performed using the Student’s *t*-test. Statistical comparisons of more than two groups were performed using one-way analysis of variance (ANOVA) with Fisher LSD post hoc tests. A *p*-value of less than 0.05 was considered to be statistically significant.

## 3. Results

### 3.1. Metastatic Colonies in Osteosarcoma Arise in Pulmonary Vasculature In Vivo

Previous findings suggest that metastatic colonies originate from pulmonary vasculature cells and, in the early stages of disease, micrometastases are contained entirely within the vasculature [25]. To clarify the mechanism involved in osteosarcoma lung metastasis, we created a metastatic lung cancer model using intravenous injections of MG63 cells. Intact organ microscopy revealed the formation of metastatic colonies in pulmonary vascular walls (Figure 1A). The establishment of metastatic colonies at distant sites is only superficially understood. One of the most important mechanisms underlying this process involves homeostatic chemokines and their receptors, which play a key role in cancer homing [26]. As the CXCL1/CXCR2 axis is required for lung metastasis in breast cancer [19], we therefore analyzed levels of CXCL1/CXCR2 expression in lung specimens. Interestingly, CXCL1 was highly expressed in pulmonary vasculature, whereas CXCR2 was only weakly stained in lung tissue. In contrast, lung metastatic foci showed strong CXCR2 expression (Figure 1A). CXCR2 expression was also determined in osteosarcoma cell lines and normal osteoblast cells. As expected, osteosarcoma cell lines expressed CXCR2 and expression levels were positively correlated with anchorage-independent growth ability, as described in a previous report [27] (Figure 1B,C). To clarify the role of CXCR2 in osteosarcoma metastasis, Transwell migration assays established MG63 sublines with high migration ability; CXCR2 expression levels were elevated in the high migration-prone sublines (MG63, M10, and M20) (Figure 1D). This evidence suggests that the CXCL1/CXCR2 axis plays a pivotal role in osteosarcoma lung metastasis. 

### 3.2. VCAM-1 Expression Is Positively Correlated with CXCR2 in Osteosarcoma Specimens 

We next examined levels of CXCR2 expression in osteosarcoma specimens, to determine the prognostic relevance of CXCR2 in osteosarcoma progression. IHC results revealed that CXCR2 expression increased with disease progression (Figure 2A). Extravasation is a critical step in metastasis, by which cancer cells are arrested in small capillaries, are extravasated, adhere to the vasculature endothelium and migrate through the vasculature wall, to establish metastatic foci. Cell adhesion molecules (CAMs) have been implicated in tumor metastasis during the extravasation process [28]. However, very little is known about CAM regulation in human osteosarcoma cells. We, therefore, examined the expression levels of VCAM-1, which has a pivotal role in tumor metastasis [29]. We found that VCAM-1 expression increased with tumor stage (Figure 2B) and was positively correlated with CXCR2 expression in osteosarcoma specimens (Figure 2C). Thus, CXCR2 expression correlates with VCAM-1 expression and tumor progression in osteosarcoma. 

### 3.3. Human Pulmonary Artery Endothelial Cell Secretion of CXCL1 Contributes to Osteosarcoma Cell Migration 

To determine whether the CXCL1/CXCR2 axis is involved in osteosarcoma lung metastasis, we examined the expression of CXCL1 in human pulmonary artery endothelial cells (HPAECs), which reside in pulmonary vasculature, where metastatic foci are found. HPAECs CM was collected and subjected to enzyme-linked immunosorbent assay (ELISA) to examine CXCL1 secretion by the HPAECs. Compared with control media, high levels of CXCL1 were found in the HPAECs CM (Figure 3A). Further testing revealed that HPAECs CM promoted migration of osteosarcoma cells, suggesting that HPAECs-secreted factor recruits osteosarcoma cells, thus contributing to homing of cancer cells (Figure 3B). This migratory ability was also seen when osteosarcoma cells were incubated with HPAECs CM in the wound healing assay (Figure 3C). To validate whether HPAECs-secreted CXCL1 plays a major role in osteosarcoma homing and migration, we used CXCL1 neutralizing antibody to block the CXCL1/CXCR2 interaction between HPAECs and osteosarcoma cells. HPAECs CM pretreated with CXCL1 antibody significantly inhibited recruitment and the migratory ability of osteosarcoma cells (Figure 3D,E). Our data show that HPAECs-secreted CXCL1 directs the homing of osteosarcoma cells to the lung, thus promoting lung metastasis in osteosarcoma.

### 3.4. HPAECs-Secreted CXCL1 Stimulates VCAM-1 Expression in Osteosarcoma Cells

In consideration of the fact that HPAECs-secreted CXCL1 recruits osteosarcoma cells and contributes to lung metastasis, and our findings showing that VCAM-1 expression is positively correlated with CXCR2 in osteosarcoma tissue, we analyzed VCAM-1 expression after HPAECs CM treatment. VCAM-1 expression levels in osteosarcoma cells were dramatically increased after treatment with HPAECs CM (Figure 4A,B). Furthermore, the recruitment of osteosarcoma cells by HPAECs CM was inhibited by transfection with VCAM-1 shRNA (Figure 4C). Preincubation of HPAECs CM with CXCL1 neutralized antibody reversed VCAM-1 expression, demonstrating that HPAECs-secreted CXCL1 can directly stimulate VCAM-1 expression in osteosarcoma cells (Figure 4D). Next, to examine whether CXCR2, the specific receptor of the ligand CXCL1, is involved in CXCL1-induced cell migration, comparison of CXCR2 among MG63, HOS, and U2OS is important. We examined the levels of CXCR2 in osteosarcoma cells by immunofluorescence, Western blotting and flow cytometry. The level of CXCR2 was significantly elevated in MG63, HOS, and U2OS cell lines (Figure 4E–G). Finally, the CXCR2 chemical inhibitor SB225002 and CXCR2 shRNA confirmed that the CXCL1/CXCR2 axis is required for VCAM-1 expression and mobility of osteosarcoma cells (Figure 4H–K).

### 3.5. The FAK/PI3K/AKT/NF-κB Signaling Cascade Is Required for HPAECs CM-Induced Increases in VCAM-1 Expression and Cell Migration

CXCR1/CXCR2 chemokine receptors elicit the PI_3_K/Akt, mitogen-activated protein kinase (MAPK) signaling cascade, in which several serine/threonine kinases are co-localized via their interaction with scaffolding proteins, in close proximity to cell-surface receptors [30]. Moreover, CXCL8/IL-8 activates protein tyrosine kinases, such as FAK and c-Src, through the CXCR2 receptor [31]. To explore which signal pathways are activated after stimulation of CXCR2 receptors with CXCL1, we screened for candidate signal pathways downstream of CXCR2. Pretreatment with pathway inhibitors (FAKi, LY294002, Wortmannin, Akti, TPCK, and PDTC) clearly inhibited cell migration and VCAM-1 expression in osteosarcoma cells (Figure 5A,B). Moreover, FAK, PI_3_K, Akt, and NF-κB were activated in response to HPAECs CM (Figure 5C,D). The dominant negative mutants of these pathway components abolished HPAECs CM-induced promotion of cell migration and VCAM-1 expression (Figure 5E,F). These findings suggest that HPAECs CM promotes cell migration and VCAM-1 through the FAK/PI_3_K/Akt/NF-κB pathway in osteosarcoma cells. 

### 3.6. The NF-κB Signaling Pathway Is Involved in HPAECs CM-Induced Increases in VCAM-1 Expression and Cell Migration

Previous research has indicated that NF-κB is an crucial transcription factor that is correlated with cancer cell migration and invasion [32]. To confirm our finding of a signaling transduction cascade, the cells were pretreated with FAK, PI_3_K, and Akt inhibitors and then incubated with HPAECs CM. The cells were subjected to Western blot to evaluate p65 phosphorylation and nuclear translocation. The inhibitors reversed HPAECs CM-induced p65 phosphorylation and nuclear translocation, confirming the involvement of the FAK/PI_3_K/Akt/NF-κB signaling cascade (Figure 6A,B). We also examined whether NF-κB transcriptional activation participates in HPAECs CM-induced increases in VCAM-1 expression. Incubation of osteosarcoma cells with HPAECs CM dramatically increased NF-κB reporter activity, in a dose-dependent manner (Figure 6C). Pretreatment with pathway inhibitors, including CXCR2, FAK, PI_3_K, and Akt, strongly inhibited NF-κB reporter activity in osteosarcoma cells incubated with HPAECs CM (Figure 6D). Knockdown of these pathway inhibitors also reduced HPAECs CM-induced promotion of NF-κB transcriptional activation (Figure 6E). Finally, the ChIP assay confirmed recruitment of p65 to the NF-κB binding element and its abolishment by pathway inhibitors (Figure 6F). These results indicate that transcriptional activation of NF-κB is required in order for HPAECs CM to induce increases in VCAM-1 expression.

## 4. Discussion

This study is the first to investigate the mechanism of lung metastasis in osteosarcoma. Human pulmonary artery endothelial cells secrete CXCL1, thus promoting the migration and homing of osteosarcoma cells to the lung. As osteosarcoma cells contain high levels of CXCR2, a chemokine receptor of CXCL1, their response to HPAECs secretion of CXCL1 promotes lung metastasis. Our investigation supports the development of CXCL1/CXCR2 as a therapeutic target to inhibit or prevent metastatic spread of disease (Figure 7).

Chemokines are small pro-inflammatory chemoattractant cytokines involved in cell activation, differentiation, adhesion and trafficking [33]. It is well known that organ-specific metastasis occurs when cancer cells are influenced by chemokine gradients at distant sites [34]. Increasing evidence indicates a pivotal role for the CXCL12/CXCR4 axis in organ-specific metastasis of various cancers [3,35,36,37]. The role of the CXCL12/CXCR4 axis in determining metastatic sites was first proposed in breast cancer [3]. Not only does the future metastatic organ express high levels of CXCL1, but its chemokine receptors CXCR4 and CCR7 are also highly expressed in human breast cancer cells. On the other hand, CXCR4/SDF-1 pathway might have a role in osteosarcoma tumor progression, supporting some of the sequential events that are involved in metastasis formation [38]. The CXCL12-CXCR4 interaction is therefore considered crucial for attracting cancer cells to a distal organ.

Recent evidence has demonstrated that tumor-secreted CXCL1 enhances tumor growth via the recruitment of various inflammatory cells [16,39] or stroma cells [40,41] into the tumor microenvironment via paracrine or autocrine mechanisms. We are the first to provide evidence showing that HPAECs-induced secretion of CXCL1 recruits CXCR2-expressing osteosarcoma cells to pre-metastatic pulmonary sites. This finding agrees with the proposed CXCL12/CXCR4 interaction in breast cancer metastasis. Clearly, chemokines and their receptors are involved in tumor metastasis, and a better understanding of chemokine signaling in this process could lead to new therapeutic strategies for cancer.

Primary tumors release cancer cells into the circulation long before diagnosis. To establish disseminated cancer cells that may eventually progress to metastases, circulating cancer cells must first transmigrate across endothelial capillary walls and then adapt to new environmental stress [42]. Transendothelial migration of monocytes is the process by which monocytes or leukocytes leave the circulatory system and extravasate through the endothelial lining of the blood vessel wall, before entering the underlying tissue [43]. Similarly to leukocyte extravasation, lung metastasis requires that cancer cells cross the lung endothelium [44]. VCAM-1 has been implicated in early leukocyte transmigration [43] and its relevance for tumorigenesis and metastasis has recently been reviewed [29]. We found that HPAECs-induced secretion of CXCL1 promoted VCAM-1 expression in osteosarcoma cells, which suggests that VCAM-1 may assist with the transendothelial migration of cancer cells. Future research should verify the role of VCAM-1 in the transendothelial migration of HPAECs.

Previous investigations have described the activation of signaling pathways stimulated by the CXCR2 receptor [30], and research has reviewed the involvement of the serine/threonine kinases, including the PI_3_K/Akt and MAPK signaling cascades [31]. HPAECs-secreted CXCL1 activates PI_3_K/Akt signaling in osteosarcoma cells. The PI_3_K/Akt onogenic pathway is critical in almost all human cancers [45], and active Akt signaling has been found by Kinome profiling in most osteosarcoma cell lines [46]. Dysregulation of this pathway is implicated in many of the pathological processes in osteosarcoma, such as tumorigenesis, proliferation, invasion, cell cycle progression, apoptosis, angiogenesis, metastasis, and chemoresistance. The evidence indicates that targeting PI3K/Akt signaling would be worthwhile in osteosarcoma metastasis. 

## 5. Conclusions

Our results provide a novel insight into osteosarcomas with lung metastases. The paracrine CXCL1/CXCR2 network links HPAECs and osteosarcoma cells, provides a metastatic trace for the cells, and directs their destination. 

## Figures and Tables

**Figure 1 cancers-12-00459-f001:**
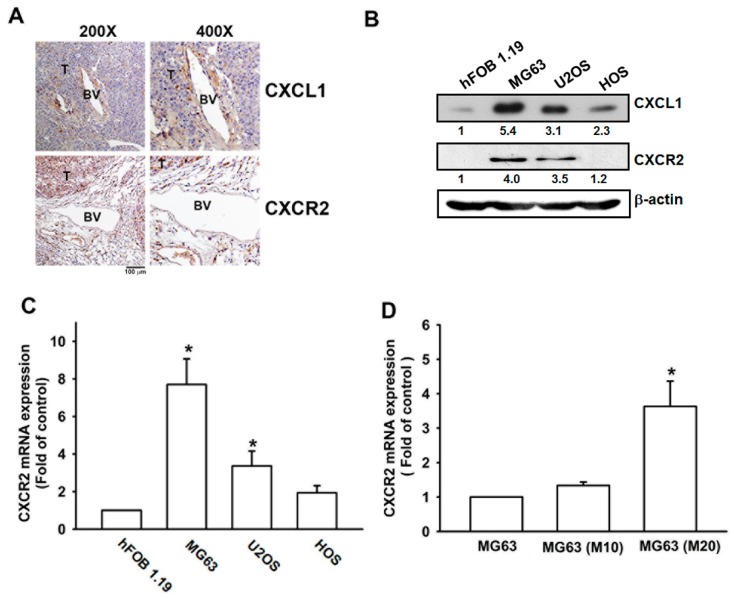
Expression of CXCL1 and CXCR2 in osteosarcoma lung metastasis. (**A**) The lung metastasis model was established by intravenous injection of MG63 osteosarcoma cells into CB17-SCID mice. Four weeks later, the lung specimens from sacrificed mice were stained with CXCL1 and CXCR2 antibodies, then photographed by optical microscope. T: tumor; BV: blood vessel. (**B**,**C**) Total cell lysate and RNA were extracted from hFOB1.19, MG63, U2OS and HOS cells, and CXCR2 expression was examined by Western blot and qPCR analysis. (**D**) The migration-prone MG63 cells were subjected to qPCR analysis to detect CXCR2 expression. Results are expressed as the mean ± SD. * *p* < 0.05 compared with the hFOB1.19 group.

**Figure 2 cancers-12-00459-f002:**
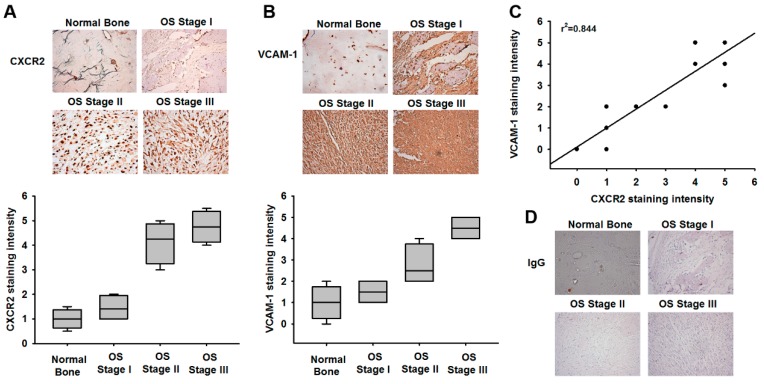
Osteosarcoma specimens exhibit significant correlations between CXCR2 and VCAM-1 expression, and tumor progression. (**A**,**B**) Tumor specimens were stained with CXCR2 and VCAM-1 antibodies, then photographed by optical microscope. The lower panels quantify the expression levels of CXCR2 and vascular cell adhesion molecule 1 (VCAM-1) in different disease stages. (**C**) Immunohistochemistry (IHC) staining scores of CXCR2 and VCAM-1 were paired from the same specimens and the correlation between CXCR2 and VCAM-1 expression levels was shown by linear regression in prostate cancer specimens. (**D**) Control IgG antibody was used as a negative control in IHC staining.

**Figure 3 cancers-12-00459-f003:**
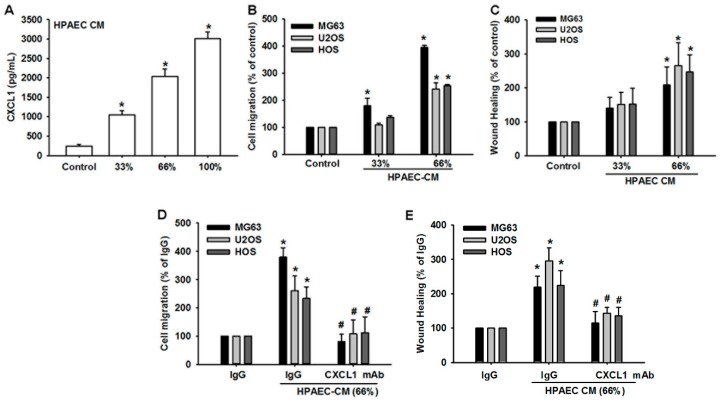
Human pulmonary artery endothelial cell (HPAECs)-secreted CXCL1 promotes migration of osteosarcoma cells. (**A**) HPAECs conditioned media (CM) was collected and levels of CXCL1 secretion were determined by ELISA. (**B**) HPAECs CM was placed in the lower chamber of the Transwell plate. MG63, U2OS, and HOS osteosarcoma cells were seeded in the upper chamber of the Transwell plate and cell mobility was determined after 20 h. (**C**) Osteosarcoma cells were incubated with the indicated concentrations of HPAECs CM for 24 h. Cell mobility was assessed by a wound healing assay. (**D**) HPAECs CM was placed in the lower chamber of the Transwell apparatus in the presence of CXCL1 neutralizing antibody or control IgG (1 μg/mL). MG63, U2OS, and HOS osteosarcoma cells were seeded in the upper chamber of the Transwell apparatus and cell mobility was determined after 20 h. (**E**) Osteosarcoma cells were incubated with the indicated concentrations of HPAECs CM in the presence of CXCL1 neutralizing antibody or control IgG for 24 h. The cells were subjected to wound healing assays to assess cell mobility. Results are expressed as the mean ± SD. * *p* < 0.05 compared with control or the control IgG group. # *p* < 0.05 compared with the HPAECs CM-treated group.

**Figure 4 cancers-12-00459-f004:**
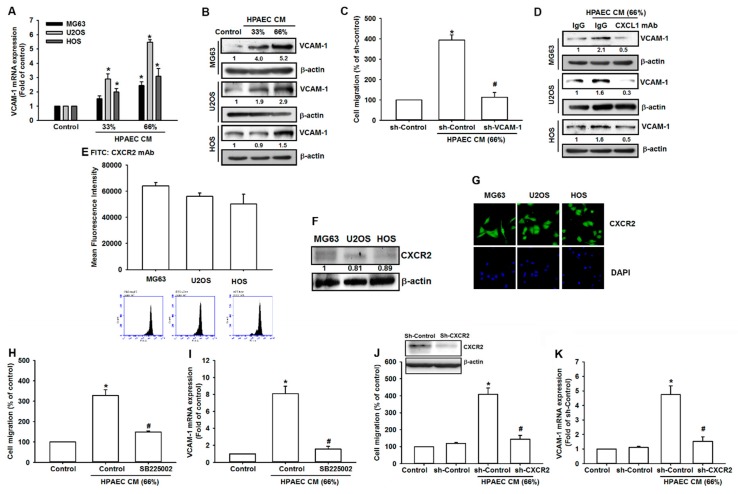
HPAECs CM promotes osteosarcoma cell mobility by upregulating VACM-1 expression. (**A**,**B**) MG63, U2OS and HOS osteosarcoma cells were incubated with increasing concentrations of HPAECs CM. Total mRNA and protein were extracted from the osteosarcoma cells and levels of VCAM-1 expression were detected by qPCR and Western blot. (**C**) MG63 cells were transfected with VCAM-1 shRNA and incubated with the indicated concentrations of HPAEs CM, and cell mobility was assessed by the Transwell migration assay. (**D**) MG63, U2OS, and HOS osteosarcoma cells were incubated with the indicated concentrations of HPAECs CM in the presence of CXCL1 neutralizing antibody or control IgG (1 μg/mL). Total protein was collected and VCAM-1 expression was evaluated by Western blot. (**E**–**G**) Total protein were collected from the indicated cell lines, and CXCR2 expression was detected using flow cytometry, Western blotting, and immunofluorescence. (**H**) MG63 cells were pretreated with the CXCR2 inhibitor SB225002 for 90 min and seeded in the upper chamber of the Transwell apparatus; then the HPAECs CM was placed in the lower chamber, and cell migration was evaluated after 20 h. (**I**) MG63 cells were pretreated with the CXCR2 inhibitor SB225002, and then incubated with HPAECs CM for 24 h. Levels of VCAM-1 mRNA expression were determined by qPCR. (**J**) MG63 cells were transfected with CXCR2 shRNA and incubated with the indicated concentrations of HPAECs CM, and cell mobility was assessed by the Transwell migration assay. (**K**) MG63 cells were transfected with CXCR2 shRNA and incubated with the indicated concentrations of HPAECs CM for 24 h, and VCAM-1 mRNA expression levels were determined by qPCR. Results are expressed as the mean ± SD. * *p* < 0.05 compared with control or the control shRNA group. # *p* < 0.05 compared with the HPAECs CM-treated group.

**Figure 5 cancers-12-00459-f005:**
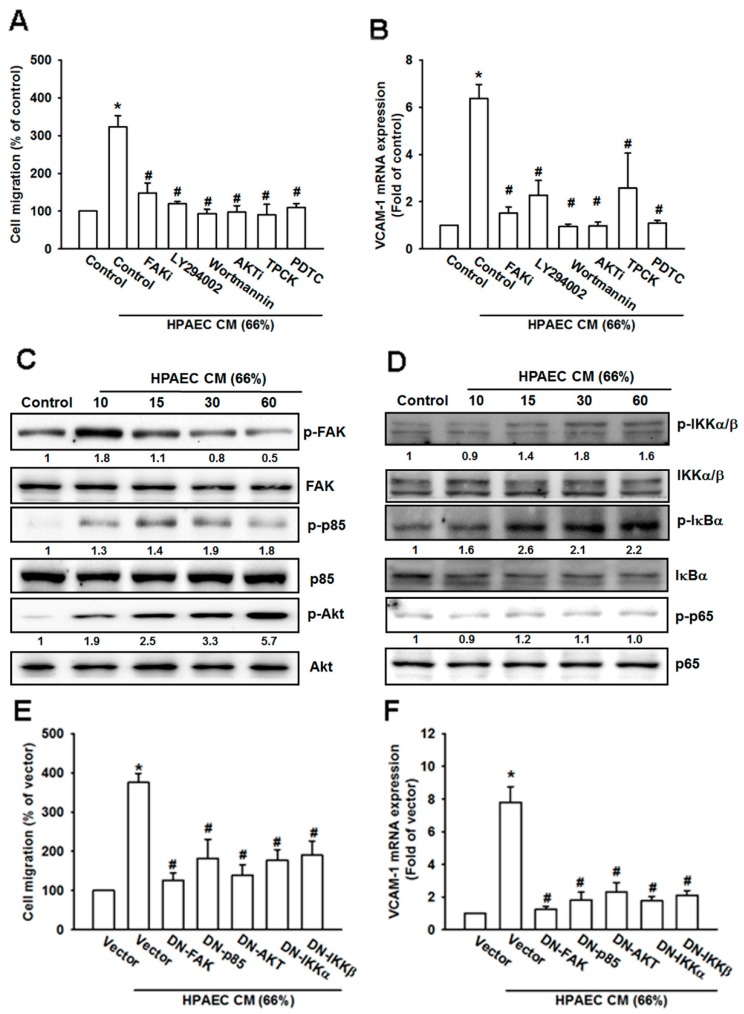
HPAECs CM increases VCAM-1 expression and cell mobility via the FAK/PI_3_K/Akt/NF-κB signaling cascade. (**A**) MG63 cells were pretreated with FAK, PI_3_K, Akt, and NF-κB inhibitors (FAKi, LY294002, Wortmanin, Akti, TPCK, and PDTC) for 90 min and seeded in the upper chamber of the Transwell plate; then the HPAECs CM was placed in the lower chamber, and cell migration was evaluated after 20 h. (**B**) MG63 cells were pretreated with FAK, PI_3_K, Akt, and NF-κB inhibitors (FAKi, LY294002, Wortmanin, Akti, TPCK and PDTC) for 90 min and then incubated with HPAECs CM for 24 h. VCAM-1 expression was investigated by qPCR. (**C**,**D**) MG63 cells were incubated with HPAECs CM for different time intervals (0, 10, 15, 30, or 60 min). Total protein was collected and phosphorylation of FAK, p85, Akt, IKKα/β, IκBα, and p65 was determined by Western blot. (**E**) MG63 cells were transfected with FAK, p85, Akt, IKKα, and IKKβ DN mutants for 24 h and then subjected to the Transwell migration assay. (**F**) MG63 cells were treated as described in (**E**) and mRNA was extracted, then VCAM-1 expression was determined by qPCR. Results are expressed as the mean ± SD. * *p* < 0.05 compared with control or the control vector group. # *p* < 0.05 compared with the HPAECs CM-treated group.

**Figure 6 cancers-12-00459-f006:**
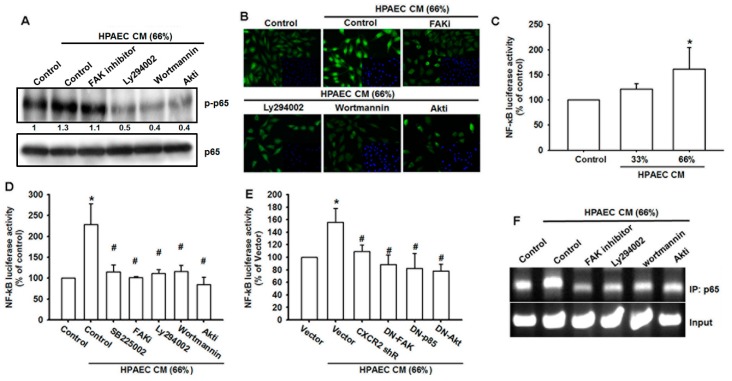
NF-κB transcriptional activation is required for cell mobility and VCAM-1 expression in osteosarcoma cells incubated with HPAECs CM. (**A**) MG63 cells were pretreated with pathway inhibitors (FAKi, LY294002, Wortmanin, Akti, TPCK, and PDTC) for 90 min and then incubated with HPAECs CM for 1 h. p65 activation was evaluated by Western blot. (**B**) MG63 cells were treated as described in (A) and then subjected to immunofluorescence by anti-p65 antibody staining. Nuclei were counterstained with DAPI (4′,6-diamidino-2-phenylindole). Representative microscopy images are shown. (**C**) MG63 cells were transfected with NF-κB reporter vector for 24 h and then incubated with different concentrations of HPAECs CM for 24 h. NF-κB transcriptional activity was examined by reporter assay. (**D**) MG63 cells were transfected with the NF-κB reporter vector for 24 h and then incubated with HPAECs CM in the presence of the indicated inhibitors for 24 h. NF-κB transcriptional activity was examined by reporter assay. (**E**) MG63 cells were co-transfected with the indicated vectors (CXCR2 shRNA, DN-FAK, DN-p85 or DN-Akt) and the NF-κB reporter vector for 24 h and then incubated with HPAECs CM for 24 h. NF-κB transcriptional activity was examined by reporter assay. (**F**) MG63 cells were treated as described in (A). Chromatin immunoprecipitation was performed with anti-p65. One percent of immunoprecipitated chromatin was assayed to verify equal loading (input). Results are expressed as the mean ± SD of triplicate samples. * *p* < 0.05 compared with control or the control vector group. # *p* < 0.05 compared with the HPAECs CM-treated group.

**Figure 7 cancers-12-00459-f007:**
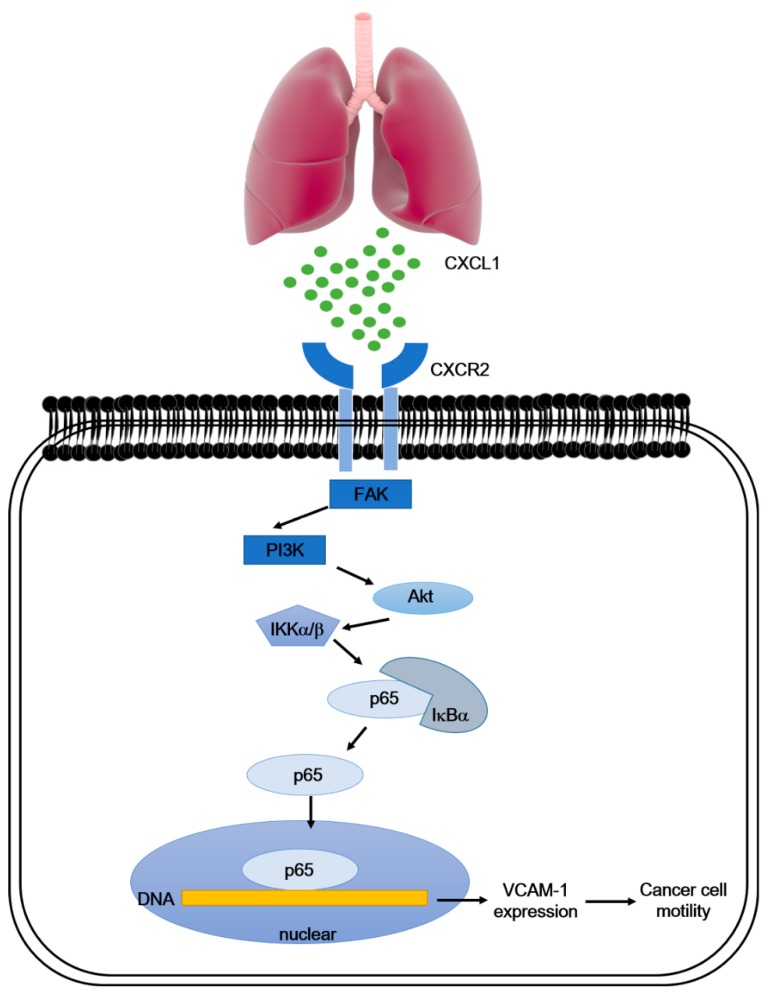
Schematic diagram summarizes the mechanism whereby CXCL 1 promotes VCAM-1 expression and cell migration in osteosarcoma. High levels of CXCL1 secreted by human pulmonary artery endothelial cells (HPAECs) promoted osteosarcoma cell mobility. When HPAECs conditioned media was incubated in osteosarcoma cells, we observed that the CXCR2 receptor and FAK/PI_3_K/Akt/NF-κB signaling cascade were required for VCAM-1 expression.

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
