# Peer review of "CXCL1/CXCR2 Paracrine Axis Contributes to Lung Metastasis in Osteosarcoma"

_cancers, 2020, doi:10.3390/cancers12020459_

Round 1
Reviewer 1 Report
The present paper describes an interesting study that demonstrates how the CXCL1/CXCR2 paracrine axis is important for developing lung metastasis in osteosarcoma. A battery of experiments both in-vitro and in-vivo have been performed to characterize the CXCL1/CXCR2 pathway.
I have the following minor comments:
1) In the methods section the M10 model of cell migration is described. However, at line 214 the M20 model is also mentioned. How this model has been obtained?
2) In paragraph 2.10 the number of mice used for the xenograft experiments is not reported.
Author Response
We are glad that the reviewers find the results of this study useful to the field and greatly appreciate their valuable and helpful suggestions.
We have prepared a new revised version of the manuscript where we take into consideration the suggestions of the academic editor and reviewers. The manuscript has been carefully revised to address the suggested aspects and a point-by-point answer to the referee’s comments follow below. We used highlight (red font) to mark the changes in Word file.
We hope that these changes will be sufficient to make our manuscript suitable for publication in Cancers.
In the methods section the M10 model of cell migration is described. However, at line 214 the M20 model is also mentioned. How this model has been obtained? Answer: We thank the reviewer for your comments. The M20 model is also provided in the methods section at line 100. In paragraph 2.10 the number of mice used for the xenograft experiments is not reported. Answer: Thanks for your suggestion. It has been reported in the revised manuscript. (Line 173). Seven animals per group, were used, and the experiment was repeated twice. For assessing lung metastasis in osteosarcoma cells in the in vivo xenograft model.

Reviewer 2 Report
Minor
Please refer to evidences regarding the CXCL12/CXCR4 axis in osteosarcoma (see Perisinotto et CCR).
Increased levels of CXCL1 are associated with increased levels are associated with tumor size … please correct
Major
… CXCR2 expression was also determined in osteosarcoma cell lines and normal osteoblast cells. As expected, osteosarcoma cell lines expressed CXCR2 and expression levels were positively correlated with anchorage-independent growth ability, as described in a previous report [27] (Fig. 1B 212 and C)…. Please clarify the picture and show that osteosarcoma cell lines (all the 3 cell lines express CXCR2 both in terms of immunohistochemistry and by cytofluorimetry.
…HPAEC CM was collected and 249 subjected to ELISA to examine CXCL1 secretion by the HPAECs. Compared with control media, high 250 levels of CXCL1 were found in the HPAEC CM (Fig. 3A). Further testing revealed that HPAEC CM 251 promoted migration of osteosarcoma cells, suggesting that HPAEC-secreted factor recruits 252 osteosarcoma cells and thus contributes to homing of cancer cells (Fig. 3B). …. These tests demonstrate that CXCL1/CXCR2 are involved in cell migration but do not show that homing is guided by CXCL1. This is a key point because other chemokines and several other factors might be implicated in pulmonary metastatic homing. Authors need to show that CXCR2 shRNA osteosarcoma cell lines abolishes lung homing preventing the interaction of CXCL1/CXCR2 axis in murine model.
Author Response
We have prepared a new revised version of the manuscript where we take into consideration the suggestions of the academic editor and reviewers. The manuscript has been carefully revised to address the suggested aspects and a point-by-point answer to the referee’s comments follow below. We used highlight (red font) to mark the changes in Word file.
We hope that these changes will be sufficient to make our manuscript suitable for publication in Cancers.
1.Please refer to evidences regarding the CXCL12/CXCR4 axis in osteosarcoma (see Perisinotto et CCR).
Answer: We thank the reviewer for your comments. I have been refered this references in the revised manuscript. (Line 393).
2.Increased levels of CXCL1 are associated with increased levels are associated with tumor size … please correct
Answer: We thank the reviewer for your comments. I have been corrected in the revised manuscript. (Line 63)
3.… CXCR2 expression was also determined in osteosarcoma cell lines and normal osteoblast cells. As expected, osteosarcoma cell lines expressed CXCR2 and expression levels were positively correlated with anchorage-independent growth ability, as described in a previous report [27] (Fig. 1B 212 and C)…. Please clarify the picture and show that osteosarcoma cell lines (all the 3 cell lines express CXCR2 both in terms of immunohistochemistry and by cytofluorimetry.
Answer: We thank the reviewer for your comments. The results of CXCR2 expression have been provided. Next, to examine whether CXCR2, the specific receptor of the ligand CXCL1, is involved in CXCL1-induced cell migration, comparison of CXCR2 among MG63, HOS and U2OS is important. We examined the levels of CXCR2 in osteosarcoma cells by immunofluorescence, Western blotting and flow cytometry. The level of CXCR2 was significantly elevated in MG63, HOS and U2OS cell lines (Figure 4E-G). (Line 284)
4. …HPAEC CM was collected and 249 subjected to ELISA to examine CXCL1 secretion by the HPAECs. Compared with control media, high 250 levels of CXCL1 were found in the HPAEC CM (Fig. 3A). Further testing revealed that HPAEC CM 251 promoted migration of osteosarcoma cells, suggesting that HPAEC-secreted factor recruits 252 osteosarcoma cells and thus contributes to homing of cancer cells (Fig. 3B). …. These tests demonstrate that CXCL1/CXCR2 are involved in cell migration but do not show that homing is guided by CXCL1. This is a key point because other chemokines and several other factors might be implicated in pulmonary metastatic homing. Authors need to show that CXCR2 shRNA osteosarcoma cell lines abolishes lung homing preventing the interaction of CXCL1/CXCR2 axis in murine model.
Answer: We thank the reviewer for your comments.
To validate whether HPAEC-secreted CXCL1 plays a major role in osteosarcoma homing and migration, we used CXCL1 neutralizing antibody to block the CXCL1/CXCR2 interaction between HPAECs and osteosarcoma cells. HPAEC CM pretreated with CXCL1 antibody significantly inhibited recruitment and the migratory ability of osteosarcoma cells (Fig. 3D and E). On the other hand, we also used the CXCR2 chemical inhibitor SB225002 and CXCR2 shRNA confirmed that the CXCL1/CXCR2 axis is required for mobility of osteosarcoma cells. Cells pretreated CXCR2 chemical inhibitor SB225002 or cells transfected CXCR2 shRNA clearly inhibited cell migration and VCAM-1 expression in osteosarcoma cells. In the further, we will investigate the mechanism of CXCR2 leading metastasis in osteosarcoma.
We really hope this revised manuscript can now be suitable for publication in “Cancers”.
Best regards,
Sincerely,
Ju-Fang Liu, PhD.

Round 2
Reviewer 2 Report
The paper addressed my concerns and is suitable for publication